# Non-linear Terahertz driving of plasma waves in layered cuprates

Francesco Gabriele[1], Mattia Udina[1] & Lara Benfatto [1✉]

The hallmark of superconductivity is the rigidity of the quantum-mechanical phase of electrons, responsible for superfluid behavior and Meissner effect. The strength of the phase stiffness is set by the Josephson coupling, which is strongly anisotropic in layered cuprates. So far, THz light pulses have been used to achieve non-linear control of the out-of-plane Josephson plasma mode, whose frequency lies in the THz range. However, the high-energy in-plane plasma mode has been considered insensitive to THz pumping. Here, we show that THz driving of both low-frequency and high-frequency plasma waves is possible via a general two-plasmon excitation mechanism. The anisotropy of the Josephson couplings leads to markedly different thermal effects for the out-of-plane and in-plane response, linking in both cases the emergence of non-linear photonics across $T_c$ to the superfluid stiffness. Our results show that THz light pulses represent a preferential knob to selectively drive phase excitations in unconventional superconductors.

[1] Department of Physics and ISC-CNR, 'Sapienza' University of Rome, Rome, Italy. ✉email: lara.benfatto@roma1.infn.it

Order and rigidity are the essential ingredients of any phase transition. In a superconductor, the order is connected to the amplitude of the complex order parameter, related to the opening of a gap $\Delta$ in the single-particle excitation spectrum. The rigidity manifests instead in the quantum-mechanical phase of the electronic wave function, associated with the phase of the order parameter[1]. Twisting the phase is equivalent to an elastic deformation in a solid, meaning that its energetic cost is vanishing for sufficiently slow spatial variations. On the other hand, as phase fluctuations come along with charge fluctuations, long-range Coulomb forces push the energetic cost of a phase gradient to the plasma energy $\omega_J$[1,2]. Although for ordinary superconductors, this energy scale is far above the THz range, in layered cuprates the existence of a weak Josephson coupling among neighboring layers[3–5] provides a natural mechanism to push down to the THz range the frequency of the interlayer Josephson plasma mode (JPM), as it was proposed long ago in order to account for the soft plasma edge appearing below $T_c$ in standard reflectivity experiments[6–10]. More recently, the possibility to manipulate such interlayer JPM by intense THz pulses has been experimentally proven[11,12], and theoretically discussed within the context of the non-linear equation of motion for the phase variable[11–16]. This approach turned out to successfully capture the main features of a series of recent experiments[17,18], even though a full quantum treatment of the JPM able to capture thermal effects across $T_c$ is still lacking. On the other hand, non-linear effects induced by strong THz pulses polarized in the planes[19–21] have been discussed so far only within the context of the SC amplitude (Higgs) mode or BCS response, that consists in lattice-modulated charge fluctuations in the clean limit[22,23]. Indeed, as their excitation energy scales in both cases as $2\Delta$, which range from 5 to 10 THz in cuprates, they appear in principle a better candidate than high-energy in-plane plasma waves. As it has been recently discussed by several authors[24–27], even small disorder affects significantly the non-linear response by triggering in general all processes mediated by the paramagnetic electronic current, that is no more conserved. This affects the relative strength of the various processes, making ultimately the Higgs mode[24–27] as well as charge/phase modes[27] dominant at strong disorder. The various processes can be further distinguished by their dependence on the pump polarization, and for cuprates the Higgs response is strongly isotropic at all disorder levels, whereas the BCS one has a shallow maximum for field polarized along the diagonal of square lattice unit cell[27]. Nonetheless, the experiments show at least two features, which do not easily match our current expectation for both the Higgs and the BCS response: (i) a monotonic temperature dependence as $T$ increases[20,21], with a persistence above $T_c$[21] and (ii) a finite and doping-dependent polarization dependence with a minimum for field polarized along the diagonal[19].

Here, we provide a complete theoretical description of the JPM contribution to the non-linear response of layered cuprate superconductors, focusing both on third-harmonic generation (THG) and pump-probe protocols for pump fields applied both out-of-plane and in plane. We first address the out-of-plane response and we show that the basic mechanism behind non-linear photonic of Josephson plasma waves is intrinsically different from the one of the Higgs mode, see Fig. 1. By pursuing the analogy with lattice vibrations in a solid, the Higgs mode is like a Raman-active optical phonon mode. It has a finite frequency at zero momentum, and its symmetry allows for a finite quadratic coupling to light[22–33]. The phase mode behaves instead like an acoustic phonon mode, pushed to the plasma energy by Coulomb interaction, carrying out a finite momentum at nonzero frequency. As such, zero-momentum light pulses can only excite simultaneously two JPMs with opposite momenta. As a consequence the excitation of out-of-plane JPMs strongly depends on the thermal probability to populate excited states and on the

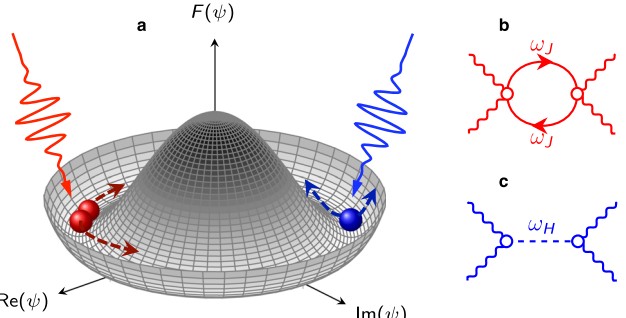

**Fig. 1 Non-linear excitation of phase and Higgs modes. a** Schematic view of the mexican-hat potential for the free energy $F(\psi)$, with $\psi$ the complex order parameter of a superconductor below $T_c$. A phase-gradient excitation corresponds to a shift along the minima, whereas a Higgs excitation moves the system away from the minimum. An intense light pulse with almost zero momentum can excite simultaneously two plasma waves with frequency $\omega_J$ and opposite momenta (in red) or a single Higgs fluctuation with frequency $\omega_H = 2\Delta$ (in blue). **b–c** Feynman-diagrams representation of the **b** plasma waves or **c** Higgs contribution to the non-linear optical response. Here wavy lines represent the e.m. field, solid/dashed lines the plasmon/Higgs field, respectively.

matching condition between the pump frequency and the JPM frequency scale, resulting in a non-monotonic dependence of the THG in temperature. We then turn our attention on the in-plane response. In this case, as the frequency scale of the in-plane JPMs is much larger than $T_c$ and of the THz pump frequency, the THG monotonically scales in temperature with the in-plane superfluid stiffness. In addition, in contrast to the Higgs mode[22,26,27], for a light pulse polarized in the planes the signal coming from JPMs is in general anisotropic, as the momenta carried out by the two plasmons can be along different crystallographic axes. All these features not only contribute to the understanding of the existing experimental measurements[17–21], but they also offer a perspective to design future experiments aimed at selectively tune non-linear photonic of Josephson plasma waves in layered cuprates.

## Results

**Two-plasmon non-linear response**. To elucidate the basic mechanism behind the two-plasmon non-linear response we first discuss the case of the out-of-plane JPM. We take a layered model with planes stacked along $z$. In the SC state the Josephson coupling $J_\perp$ of the SC phase $\phi_n$ between neighboring planes sets an effective XY model:

$$H = -J_\perp \sum_n \cos(\phi_n - \phi_{n+1}). \tag{1}$$

An electric field polarized along $z$ enters the Hamiltonian via the minimal-coupling substitution[1] $\theta_n \to \theta_n - (2\pi/\Phi_0)dA_z$, with $\theta_n = \phi_n - \phi_{n+1}$, $d$ interlayer distance and $\Phi_0 = hc/(2e)$. The corresponding out-of-plane current density $I_z = -\partial H/\partial(cA_z)$ is given by:

$$I_z = J_c \sin(\phi_n(t) - \phi_{n+1}(t) - (2\pi/\Phi_0)dA_z(t)), \tag{2}$$

where $J_c = 2eJ_\perp/\hbar S$, with $S$ surface of each plane. The Josephson current (2) naturally admits an expansion in powers of $A_z$ to all orders:

$$\langle I_z \rangle = \chi_z^{(1)} A_z + \chi_z^{(3)} A_z^3 + \cdots, \tag{3}$$

where the explicit time convolution of Eq. (3) has been omitted for compactness. Here, following the same approach used so far to investigate the Higgs response[22,23,28,29], we rely on a quasi-equilibrium description, where the leading effect of the intense THz pump field is to trigger a third-order $\chi^{(3)}$ response mediated by

plasma waves. The quantum generalization of the model (1) has been widely discussed within several contexts[11,14,15,32,34,35]. Here we follow the approach of refs. [34,35] where long-range Coulomb interactions are introduced within a layered model appropriate for cuprates (see Methods). The Gaussian quantum action for the phase mode at long wavelength has the usual form:

$$S_\perp^G \simeq \frac{\nu S}{2d} \sum_{i\omega_m,k_z} 4\sin^2(k_z d/2)\left[\omega_m^2 + \omega_J^2\right]|\phi(i\omega_m, k_z)|^2, \quad (4)$$

where $\omega_J^2 = c^2/\varepsilon\lambda_c^2 = 8\pi e d J_c/\hbar\varepsilon$ is the energy scale of the out-of-plane JPM, $i\omega_m = 2\pi m T$ are Matsubara frequencies and $\nu = \hbar^2\varepsilon/(16\pi e^2)$, with $\varepsilon$ the background dielectric constant. In the classical limit only $\omega_m = 0$ is relevant and one recovers the leading term of Eq. (1), i.e., a discrete phase gradient along $z$, as expected for the Goldstone mode.

To compute the third-order contribution in Eq. (3) we need to derive the effective action $S^{(4)}$ for the gauge field up to terms of order $\mathcal{O}(A_z^4)$ (see Methods). By coupling the gauge field $A_z$ to the phase mode via the minimal-coupling substitution in Eq. (2) and by expanding the cosine term, one finds that:

$$S = S_\perp^G + \frac{\pi^2 J_\perp}{\Phi_0^2} \sum_{n,\mathbf{r}_i} \int d\tau A_z^2(\tau)\theta_{n,\mathbf{r}_i}^2(\tau) + \cdots, \quad (5)$$

where dots denote additional terms not relevant for the $\chi^{(3)}$ response. The second term in Eq. (5) can be treated as a perturbation with respect to $S_\perp^G$, see Supplementary Note 2, so that integrating out the JPM one obtains:

$$\begin{aligned} S_\mathbf{A}^{(4)} &= \int d\tau \int d\tau' A_z^2(\tau) K_\perp(\tau - \tau') A_z^2(\tau') \\ &= \sum_{i\omega_m} A_z^2(i\omega_m) K_\perp(i\omega_m) A_z^2(-i\omega_m), \end{aligned} \quad (6)$$

where $A_z^2(i\omega_m)$ is defined as the Fourier transform of $A_z^2(\tau)$ and $K_\perp(i\omega_m)$ is the non-linear optical kernel of the system, given by the convolution of two JPM propagators, as represented diagrammatically in Fig. 1b. After analytical continuation to real frequency we get:

$$K_\perp(\omega) = K_0 \frac{J_\perp^2}{\omega_J} \frac{\coth(\beta\omega_J/2)}{4\omega_J^2 - (\omega + i\gamma)^2}, \quad (7)$$

with $K_0$ a constant prefactor and $\gamma$ accounts for the plasmon dissipation (see Supplementary Note 1 and 2). From Eq. (6), it immediately follows (see Methods) that $\langle I_z^{NL}(t)\rangle = 4\int dt' A_z(t) K_\perp(t - t') A_z^2(t')$. Therefore, for a monochromatic incident field $A_z = A_0\cos(\omega t)$ the non-linear current admits both a term oscillating at $\omega$, which gets mixed with the linear response, and one oscillating at $3\omega$, whose intensity is given by[22,23,28,29]

$$I^{THG} = I_0|K(2\omega)|^2, \quad (8)$$

where $I_0$ is an overall constant. The vanishing of the denominator in Eq. (7) identifies the resonance of the non-linear kernel. As the physical mechanism behind the THG is the excitation of two plasma waves, the largest $I^{THG}$ in Eq. (8) occurs when twice the pump frequency matches the $2\omega_J$ kernel resonance, i.e., $\omega = \omega_J$. This has to be contrasted, e.g., to the case of the THG from the Higgs mode. In this case, the electromagnetic (e.m.) field excites non-linearly a single amplitude fluctuation $\delta\Delta$, via a term like $A^2\delta\Delta$[22,23,28,29]. As a consequence the non-linear kernel, identified by the dashed line in Fig. 1c, is proportional to a single Higgs fluctuation, and the THG (8) is resonant when the pump frequency matches half the mode energy, i.e., when $\omega = \omega_H/2 = \Delta$, as observed in conventional superconductors[28,31] for strong (up to $\sim$100 kV/cm) but not too intense fields[36].

**Out-of-plane THG.** Once derived the two-plasmon contribution to the non-linear optical kernel, let us compute the THG for a field polarized in the out-of-plane direction. The temperature dependence of the JPM non-linear kernel (7) and the corresponding THG (8) for a narrow-band pulse are shown in Fig. 2a–d for different values of the pump frequency $\omega$. Here we modeled $J_\perp(T)$ and the corresponding $\omega_J(T)$ according to the out-of-plane superfluid stiffness measured in ref. [18]. In general, the THG for the out-of-plane response is not monotonic, as one has to face with three different temperature effects in $K(2\omega)$: (i) the suppression of $J_\perp(T)$ and $\omega_J(T)$ with temperature; (ii) the increase of $\coth(\beta\omega_J)$ with temperature, owing to thermal activation of the plasmon population; (iii) the resonance condition $2\omega = 2\omega_J(T)$, that is achieved at the temperature where the (fixed) pump frequency matches the value of $\omega_J$, and depends on the relative value of the pump frequency $\omega$ with respect to $\omega_J(T = 0) \equiv \omega_{J,0}$. In the case where $\omega < \omega_{J,0}$, as for $\omega = \omega_3$ in Fig. 2a, the temperature dependence of $I^{THG}(\omega_3)$ is dominated by the maximum at the temperature where $\omega_J(T) = \omega_3$. On the other hand, when $\omega \geq \omega_{J,0}$, as it is the case for $\omega = \omega_1, \omega_2$, the resonant excitation of the plasma mode cannot occur, and the temperature dependence of the THG is controlled by the opposite effects (i–ii), which lead to a non-monotonic dependence of $I^{THG}(T)$. The thermal effect (ii) is particularly pronounced for the out-of-plane JPM, as $\omega_{J,0}$ is of the same order of the critical temperature $T_c$. The absolute value of $I^{THG}$ depends also on the damping $\gamma$ present in Eq. (7), which has the same role of a linear damping term in the equations-of-motion approach, see Supplementary Note 1. In Fig. 2c, d, we show the results for a temperature-dependent $\gamma(T) = \gamma_0 + r(T)$, where $r(T) = r_0 e^{-\Delta/T}$ has been taken in analogy with previous work[16] to mimics dissipative effects from normal quasiparticles. In this case the plasma resonance is progressively smeared out by increasing temperature, and for out-of-resonance conditions, the THG signal rapidly loses intensity as the system is warmed up.

The THG for a field polarized along $z$ has been measured so far only by means of a broadband pump[18]. To make a closer connection with this experimental setup we then simulated (see Methods) the THG for a short ($\tau = 0.85$ ps) pump pulse $E_p(t)$ with central frequency $\Omega/2\pi = 0.45$ THz, as shown in Fig. 2g. The frequency spectrum of the resulting non-linear current $I_z^{NL}$ presents then a broad peak around $3\Omega$, as shown in Fig. 2e. The integrated spectral weight of the $3\Omega$ peak is shown in Fig. 2f at several temperatures. Following ref. [18] we used $\Omega \simeq \omega_{J,0}$, so the narrow-band response should corresponds to the case $\omega = \omega_2$ of Fig. 2d. However, the broadband spectrum of the pump pulse enhances the response at intermediate temperatures and apart from a small deep around $T = 0.2T_c$ the signal scales with the superfluid stiffness, in good agreement with the available experimental data.

**Out-of-plane pump-probe oscillations.** In the broadband case, the nature of the non-linear kernel can also be probed via a typical pump-probe experimental setup, schematically summarized in Fig. 2g. As it has been theoretically described in ref. [23,37] for the transmission geometry, the oscillations of the differential probe field with and without the pump $\delta E_{pr}(t_{pp})$ as a function of the pump-probe time delay can be directly linked to the resonant non-linear optical kernel. In the case of the out-of-plane response (7) one then obtains (see Methods):

$$\begin{aligned} \delta E_{pr}(t_{pp}) &\propto \int dt K(t_{pp} - t) A_z^2(t) \\ &= F(T)\int_{-\infty}^{t_{pp}} e^{-\gamma(t - t_{pp})} \sin(2\omega_J(t_{pp} - t)) A_z^2(t) \end{aligned} \quad (9)$$

where $F(T) \equiv J_\perp^2 \coth(\beta\omega_J/2)/\omega_J^2$. When the pump pulse is short enough one can approximate $A_z^2(t) \simeq \delta(t)$ and Eq. (9) shows that the differential field $\delta E_{pr}(t_{pp})$ oscillates at twice the JPM frequency,

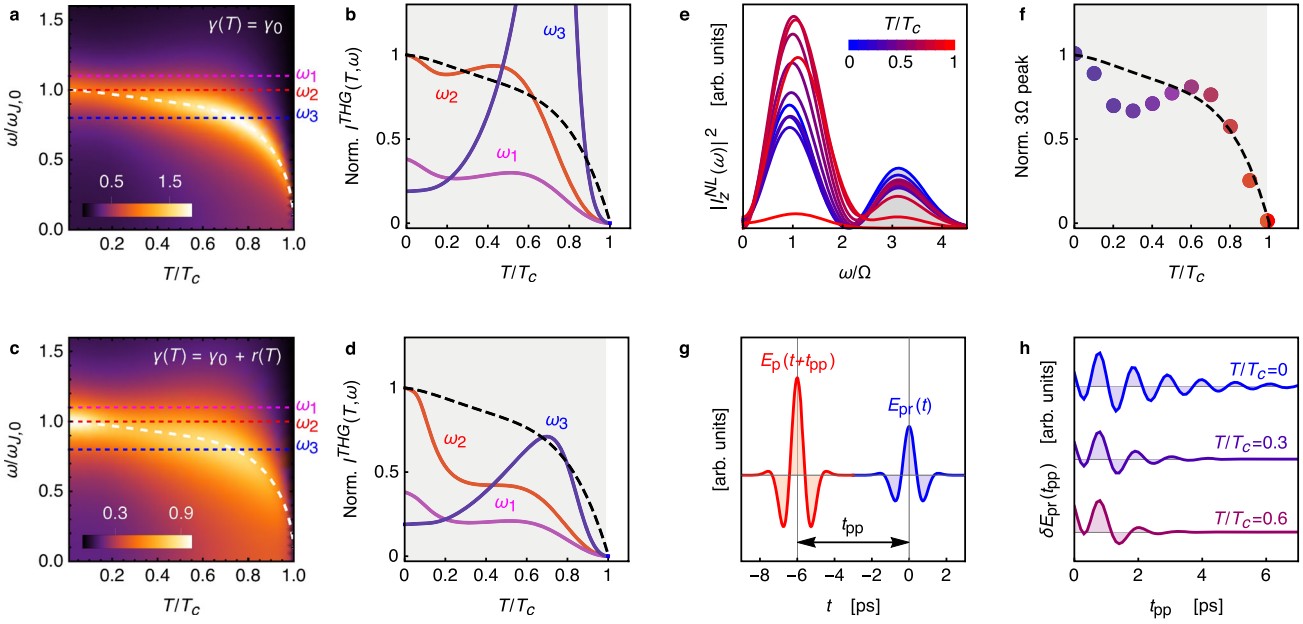

**Fig. 2 Non-linear excitation of out-of-plane JPM. a–d** Narrow-band pulse. Temperature and frequency dependence of the non-linear kernel (7) $|K(2\omega, T)|$, normalized to its $T = 0$ value for the pumping frequency $\omega = \omega_{J,0}$, for constant **a** and temperature-varying **c** damping $\gamma$. The dashed line denotes $\omega_J(T)/\omega_{J,0}$. **b, d** show the corresponding $I^{THG}(\omega_i, T)$ for three values $\omega_i$ of the pump frequency, normalized to its $T = 0$ value for the pumping frequency $\omega_2 = \omega_{J,0}$. The dashed line represents $J_\perp(T)/J_\perp(0)$. **e–h** Broadband pulse. **e** Spectrum of the non-linear current $I_z^{NL}$ as a function of frequency, normalized to the central frequency $\Omega$ of the pump pulse, shown explicitly in **g**. The intensity of the THG signal is now obtained by integrating the peak around $3\Omega$ (gray region in **e**). Its temperature dependence, normalized to its $T = 0$ value, is shown in **f**, with the same color code of the curves of **e**. **g** Schematic of the pump-probe setup: a weak probe field $E_{pr}(t)$ impinges on the sample with a variable time delay $t_{pp}$ with respect to the intense pump pulse $E_p(t)$. **g** Time-dependence of the differential probe field $\delta E_{pr}(t_{pp})$ measured with and without the pump, at different temperatures. The periodicity of the oscillations matches the $2\omega_J(T)$ value at each temperature. Here we set $\omega_{J,0}/2\pi = 0.47$ THz in accordance with the experiments[17].

and not at the frequency of the mode, as it occurs for the Higgs mode observed in conventional superconductors[38]. This prediction is confirmed when a realistic pump pulse is used in Eq. (9), as shown in Fig. 2h, which reproduces very well the $2\omega_J$ oscillations reported at low-temperature in pump-probe experiments in reflection geometry[17].

**In-plane THG.** Let us consider now the effects of a strong THz pulse polarized within the plane. In this case, we can generalize the model (4) by taking into account both the two-dimensional nature of the phase fluctuations in the plane and the anisotropy of penetration depth measured experimentally in cuprates[3–5], where $\lambda_c \simeq 10-100\lambda_{ab}$ depending on the material and the doping, and $\lambda_{ab} \simeq 2000$ Å, so that $\omega_J^\parallel = c/\sqrt{\varepsilon}\lambda_{ab}$ is much larger than the out-of-plane one. Following again the microscopic derivation outlined, e.g., in ref. [34,35] we obtain

$$S_\parallel^G \simeq \frac{\nu}{2} \sum_{i\omega_m, \mathbf{k}} \mathbf{k}^2 \left[ \omega_m^2 + (\omega_J^\parallel)^2 \right] |\phi(i\omega_m, \mathbf{k})|^2, \quad (10)$$

where $\mathbf{k} = (k_x, k_y)$ and we promoted the phase difference to a continuum gradient for the in-plane phase mode. To describe the non-linear coupling to the e.m. field, we rely again on a quantum $XY$ model, whose coupling constant is the effective in-plane stiffness $J_\parallel = \hbar^2 c^2 d / 16\pi e^2 \lambda_{ab}^2$. Even though the microscopically derived phase-only action is not in general equivalent to the $XY$ model[35], for cuprates this can still represent a reasonable starting point[34]. By minimal-coupling substitution $\nabla \phi(\mathbf{r}) - (2\pi/\Phi_0)\mathbf{A}_\parallel$ we then obtain, in full analogy with Eq. (5), that:

$$S = S_\parallel^G + \frac{J_\parallel}{4!} \int d\mathbf{r}d\tau \left[ A_x^2(\tau)(\partial_x\phi)^2 + A_y^2(\tau)(\partial_y\phi)^2 \right] + \cdots. \quad (11)$$

By following the same steps as before we obtain a quartic action

of the form (6), but the non-linear kernel becomes a tensor, which admits two different $K_{xx;xx}$ and $K_{xx;yy}$ components (see Methods):

$$K_{xx;xx} = 3K_\parallel, \quad K_{xx;yy} = K_\parallel \quad (12)$$

where $K_\parallel$ has the same structure of Eq. (7), provided that $J_\perp$ and $\omega_J$ are replaced by $J_\parallel$ and $\omega_J^\parallel$. The frequency and temperature dependence of $K_\parallel$ is shown in Fig. 3a. The in-plane stiffness $J_\parallel$ is taken as linearly decreasing, in analogy with experiments[3–5]. As $\omega_J^\parallel(T = 0) \equiv \omega_{J,0}^\parallel$ is of the order of the eV, we only considered the case of THz pump frequencies $\omega_i < \omega_{J,0}^\parallel$. As one can see, when $\omega_i$ is a fraction of $\omega_{J,0}$ the resonance condition $\omega_i = \omega_J^\parallel(T)$ is still attained at temperatures where the kernel is large enough to give rise to a pronounced maximum in the THG intensity. However, when $\omega_i \ll \omega_{J,0}^\parallel$ the resonance is only attained near to $T_c$ where the prefactor has already washed out the two-plasmon resonance, and the THG scales with the superfluid stiffness. This is easily seen from Eq. (7), since by putting $\omega \simeq 0$ in the denominator, and considering that $\coth(\beta\omega_J^\parallel) \simeq 1$ at all relevant temperatures, from $\omega_J^\parallel \propto \sqrt{J_\parallel}$ one finds

$$I^{THG}(T, \omega \ll \omega_{J,0}^\parallel) \sim J^\parallel(T). \quad (13)$$

The scaling of the THG intensity in the THz regime with $J_\parallel$ has several consequences. First, $I^{THG}$ monotonically increases below $T_c$, in striking contrast with the pronounced maximum one would expect for resonance at $\omega_i = \Delta(T)$, owing to the Higgs[28,29] or BCS response[22–25,27]. Second, the superfluid stiffness appearing in the THG response is the one measured at THz frequencies. As such, owing to both fluctuations effects and inhomogeneity it vanishes in cuprates well above $T_c$[21,39,40]. Interestingly, whatever is the origin of persistence of the finite-frequency stiffness above $T_c$, it directly

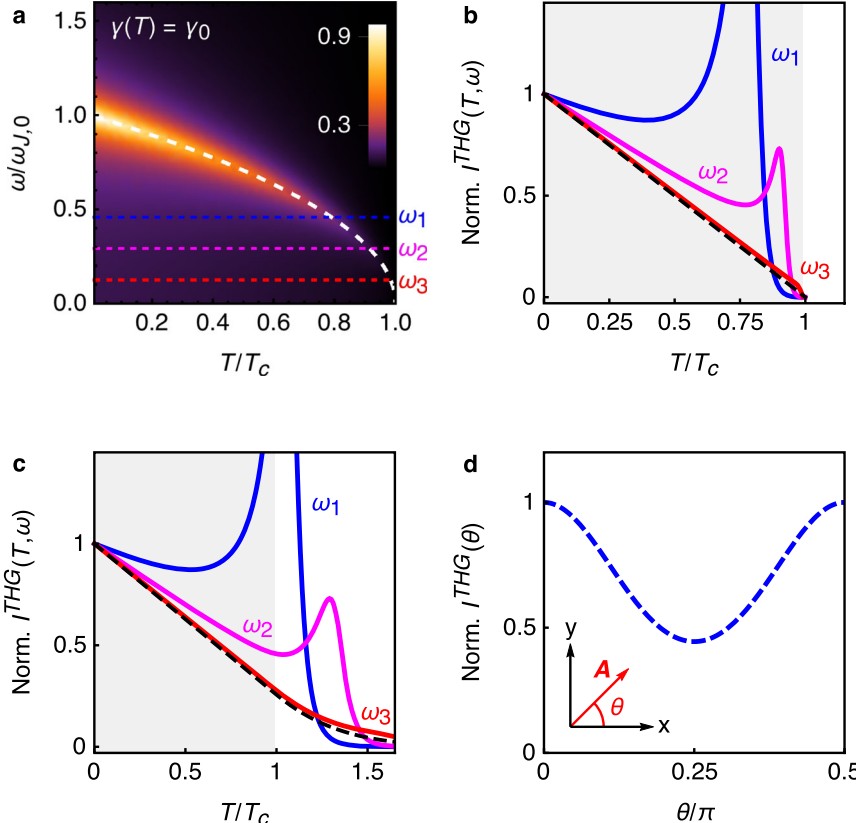

**Fig. 3 Non-linear excitation of in-plane JPM. a** Temperature and frequency dependence of the non-linear kernel $|K_\parallel(2\omega, T)|$, normalized to its $T=0$ value for the pumping frequency $\omega = \omega_{J,0}$, for constant damping $\gamma$. The dashed line denotes $\omega_\parallel^\parallel(T)/\omega_{J,0}^\parallel$. **b** $I^{THG}(\omega_i, T)$ for three $\omega_i$ values of the pump frequency, marked in **a**, normalized to $I^{THG}(\omega_i, 0)$. The dashed line represents $J_\parallel(T)/J_\parallel(0)$. As one can see, when $\omega_{J,0}^\parallel/\omega_i$ is increased the THG intensity progressively approaches the temperature dependence of the stiffness. **c** Effect of superconducting fluctuations on the THG. Here the dashed line simulates the experimental behavior of the $J_\parallel(T)$ measured at THz frequencies, with a pronounced tail above $T_c$. When $\omega_{J,0}^\parallel/\omega_i$ is increased also the THG signal survives above $T_c$, following the fluctuating stiffness. **d** Angular dependence of $I^{THG}(\theta) = |K(\theta)|^2$, where $K(\theta)$ is given by Eq. (14).

implies a persistence of the $I^{THG}$ above $T_c$, as we exemplify in Fig. 3c, where we report a simulation of the superfluid stiffness with a finite tail above $T_c$ (see also Supplementary Note 3 for more details). Both the monotonic suppression[20] and the persistence of non-linear effects above $T_c$[20,21] have been recently reported in THG and THz Kerr measurements in cuprate superconductors. As explained above, they can hardly be reconciled with the typical $2\Delta$ resonance expected for the BCS response or for the Higgs mode, both within clean[22,23] and disordered[24,25,27] models for superconductors. A second experimental finding that does not properly fit the BCS and Higgs scenario for the in-plane pump field is the polarization dependence of the response, i.e., the dependence of $I^{THG}$ on the angle $\theta$, which the pump field forms with the $x$ crystallographic axis. Indeed, within a disordered superconducting model with a realistic band structure, the Higgs signal has an isotropic contribution, while the BCS one has a relative maximum for a field applied along the diagonal[27]. As a consequence, the recent observation[19] of a sizeable response with a minimum along the diagonal direction in optimal and over-doped Bi2212 compounds cannot be simply ascribed to these collective excitations. It is then worth exploring the polarization dependence of the JPM signal. Owing to the tensor structure of the in-plane kernel (12), the non-linear current owing to JPM for a pump field with a polarization angle $\theta$ scales with:

$$K(\theta) = K_{A_{1g}} + K_{B_{1g}} \cos^2(2\theta) \qquad (14)$$

where we introduce the standard decomposition of the kernel by means of the irreducible representation of the square lattice, i.e.,

$K_{A_{1g}/B_{1g}} = (K_{xx;xx} \pm K_{xx;yy})/2$. The resulting $I^{THG}(\theta) \propto |K(\theta)|^2$ is shown in Fig. 3d. According to Eq. (12), for JPM is $K_{A_{1g}}/K_{B_{1g}} = 2$. As mentioned above, so far JPM are the only candidate to give a $B_{1g}$ contribution to the THG. In this view, the doping dependence of the anisotropy observed experimentally within the Bi2212 family[19], and the reported cuprate-family dependence[20], both offer a potentially privileged knob to explore the relative importance of phase-fluctuation effects in cuprates. It is worth noting that the anisotropy of the kernel for JPMs follows form the intrinsics anisotropy of the two-plasmon excitation process, and the ratio $K_{A_{1g}}/K_{B_{1g}} = 2$ only holds within the phenomenological approach based on the quantum $XY$ model, where the overall coupling constant of the action (11) is isotropic. However, within a microscopically derived phase-only model the interacting terms in the phase can differ from the one obtained within the $XY$ model, as discussed for the clean case in ref. [35]. As a consequence, although one expects, in general, an anisotropy of the non-linear JPM response, the value of the $K_{A_{1g}}/K_{B_{1g}}$ ratio could also be influenced by microscopic details.

**Discussion**. Our work establishes the theoretical framework to manipulate and detect JPMs in layered cuprates across the superconducting phase transition. The basic underlying mechanism relies on the excitation of two plasma waves with opposite momenta by an intense field. For the out-of-plane response, we support the well-established approach based on non-linear sine-Gordon equations[11,14,15,17,18], adding a complete

description of thermal effects and highlighting the possibility to tune the resonant excitation of JPMs by changing the temperature. For the in-plane response, we suggest the possible relevance of JPMs to explain several puzzling aspects emerging in recent measurements in different families of cuprates[19–21]. Although for the out-of-plane response the strong incoherent quasiparticle transport automatically suppresses all electronic mechanisms, leaving the JPM as the only plausible candidate to explain non-linear effects, for the in-plane case an open question remains a quantitative estimate of the signal coming from the JPMs, as compared with the one owing to the Higgs or to BCS quasi-particle excitations. Indeed, as the recent theoretical work on disordered superconducting models demonstrated[24–27], even weak disorder becomes crucial to estimate the relative strength of the various possible processes, and to establish the polarization dependence of the response[27]. Nonetheless, as we have shown, even in the absence of a quantitative estimate of the hierarchy of the various effects the temperature and polarization dependence of the non-linear response can be used to discriminate different contributions. In our modeling, the large value of the in-plane plasma frequency comes along with a large value for the in-plane stiffness $J_\parallel$, which controls the non-linear coupling of the JPM to the e.m. field. This suggests that especially near optimal doping, where $J_\parallel$ attains its maximum value, a two-plasmon THG signal can be comparable to other effects. An interesting additional question is a possibility that a finite supercurrent triggered by a very strong THz field, as the one recently discussed within the context of second-harmonic generation in conventional super-conductors[31,41], could also allow for single-plasmon excitations processes. From this perspective, the theoretical and experimental investigation of non-linear phenomena induced by intense THz pulses represents a privileged knob to probe the relative strength of pairing and phase degrees of freedom in unconventional superconducting cuprates.

## Methods

**Effective quantum action**. The derivation of the quantum action for the phase degrees of freedom can be done following a rather standard approach, see, e.g., refs. [1,34,35] and references therein. The basic formalism relies on the quantum action representation of a microscopic superconducting model in the presence of long-range Coulomb interactions. The collective variables corresponding to the amplitude, phase, and density degrees of freedom are introduced via an Hubbard–Stratonovich decoupling of the interacting superconducting and Cou-lomb term. This allows one to integrate out explicitly the fermionic degrees of freedom in order to obtain a quantum action in the collective variables only, whose coefficients are expressed in terms of fermionic susceptibilities, computed on the SC ground state. The result for the Gaussian phase-only action in the isotropic three-dimensional case reads:

$$S_{eff}[\theta] = \frac{1}{8}\sum_{i\omega_m,\mathbf{q}}\left[\hbar^2\omega_m^2\tilde{\chi}_{\rho\rho} + D_s\mathbf{q}^2\right]|\phi(i\omega_m,\mathbf{q})|^2. \tag{15}$$

Here $D_s = \hbar^2c^2/4\pi e^2\lambda^2$ and $\tilde{\chi}_{\rho\rho}$ is the density–density susceptibility dressed at RPA level by the Coulomb interaction $V(\mathbf{q})$:

$$\tilde{\chi}_{\rho\rho} = \frac{\chi^0_{\rho\rho}}{1 + V(\mathbf{q})\chi^0_{\rho\rho}}, \tag{16}$$

where $\chi^0_{\rho\rho}$ represents the bare charge susceptibility, which reduces in the static limit to the compressibility of the electron gas, i.e. $\chi^0_{\rho\rho}(\omega_n = 0, \mathbf{q} \to 0) \equiv \kappa$. The nature of the Goldstone phase mode is dictated by the form of the charge susceptibility. For the neutral system, Coulomb interactions are absent and $\tilde{\chi}_{\rho\rho}$ in Eq. (15) can be replaced by the bare one $\chi^0_{\rho\rho}$. Thus, in the long-wavelength limit the pole of the Gaussian phase propagator defines, after analytical continuation to real frequencies $i\omega_n \to \omega + i\delta$, a sound-like Goldstone mode: $\omega^2 = (D_s/\kappa)\mathbf{q}^2$. On the other hand, in the presence of Coulomb interaction the long-wavelength limit of the charge compressibility (16) scales as $\tilde{\chi}_{\rho\rho} \to 1/V(\mathbf{q})$. In the usual isotropic three-dimensional case $V(\mathbf{q}) = 4\pi e^2/\mathbf{q}^2$, where $\varepsilon$ is the background dielectric constant, and one easily recovers from Eq. (15) that

$$S_{eff}[\phi] = \frac{1}{2}\sum_{i\omega_m,\mathbf{q}}\frac{\hbar^2}{4V(\mathbf{q})}\left[\omega_m^2 + \omega_P^2\right]|\phi(i\omega_m,\mathbf{q})|^2, \tag{17}$$

where $\omega_P^2 \equiv 4\pi e^2 D_s/\hbar^2\varepsilon = c^2/\lambda^2\varepsilon$ coincides with the usual 3D plasma frequency. In the case of cuprates, one should start from a layered model where the in-plane and out-of-plane superfluid densities are anisotropic, so that the $D_s\mathbf{q}^2$ term in Eq. (15) is replaced by $(4D_\perp/d^2)\sin^2(k_z d/2) + D_\parallel k_\parallel^2$, with $D_{\perp/\parallel} = \hbar^2c^2/4\pi e^2\lambda^2_{c/ac}$. In addition, one can also introduce an anisotropic expression for the Coulomb interaction, to account for the discretization along the $z$ direction[34]. Following, e.g., the derivation of ref. [34] one then recovers in the long-wavelength limit the two expressions (4) and (10). Notice that at long-wavelengths the result (4) coincides also with the one based on the non-linear sine-Gordon equations, as shown in refs. [11,14,15]. In this case, however the effect of long-range forces is included via the coupling to the electromagnetic gauge and scalar potentials, which are eliminated to derive the equations of motion for the phase variables. Further technical details on this analogy are provided in Supplementary Note 1.

**Computation of the non-linear kernel**. The current $I_\alpha$ in the $\alpha = (x, y, z)$ direction is defined as usual as the functional derivative with respect to $A_\alpha$ of the action $S_A$. Thus, to compute the third-order contribution to $I_z$ in Eq. (3) we need to expand the e.m. action up to terms of order $\mathcal{O}(A_z^4)$. The coupling term of the JPM to $A_z^2$ in Eq. (5) leads to a $A_z^4$ contribution after integrating out the plasmon, see Supplementary Note 2. This is represented by the Feynmann diagram of Fig. 1b. Here each solid line denotes the Gaussian phase mode, obtained by Eq. (4) as $\langle|\phi(i\omega_m,k_z)|^2\rangle = \left\{4\sin^2(k_z d/2)\left[\omega_m^2 + \omega_J^2\right]\right\}^{-1}$. With straightforward calculations one gets:

$$S_A^{(4)} = K_0\sum_{i\omega_m}T\sum_{i\omega'_m}\frac{A_z^2(i\omega_m)J_\perp^2 A_z^2(-i\omega_m)}{[(\omega_m + \omega'_m)^2 + \omega_J^2][\omega'^2_m + \omega_J^2]}, \tag{18}$$

where $A_z^2(i\omega_m) = \sum_{i\omega'_m}A_z(i\omega'_m)A_z(i\omega_m - i\omega'_m)$ is the Fourier transform on $A_z^2(\tau)$. Eq. (18) coincides with Eq. (6), once defined $K_\perp(i\omega_m) = K_0\frac{J_\perp^2}{\omega_J}\frac{\coth(\beta\omega_J/2)}{4\omega_J^2 + \omega_m^2}$. After analytical continuation $i\omega_m \to \omega + i\delta$ to real frequencies one then recovers Eq. (7). From Eq. (18) one directly derives $I_z(i\omega_n) = -\partial S_A^{(4)}/\partial A_z(-i\omega_n) = -4\sum_{i\omega'_m}A_z(i\omega_n - i\omega'_m)K(i\omega'_m)A_z^2(i\omega'_m)$, we used the parity of the kernel to write the four possible derivatives in the same way. After analytical continuation to real frequency one has:

$$I_z^{NL}(\omega) = 4\int d\omega' A_z(\omega - \omega')K_\perp(\omega')A_z^2(\omega'), \tag{19}$$

whose Fourier transform to real time gives $\langle I_z^{NL}(t)\rangle = -4\int dt' A_z(t)K_\perp(t - t')A_z^2(t')$, as stated in the main text. For a monochromatic field, $A(\omega)$ is proportional to a delta function peaked at the pump frequency, and one recovers Eq. (8). Notice that the infinitesimal positive $\delta$ in the analytical continuation of the kernel is promoted here to a finite and temperature-dependent value $\gamma(T) = \gamma_0 + r_0e^{-\Delta/T}$ to account for plasmon dissipative effects, as explained in Supplementary Note 1. To better reproduce the pump-probe experimental findings[17], in Fig. 2 we fixed $\gamma_0/2\pi = 0.08$ THz, while $r_0 = 0.3\omega_{J,0}$ in panels c,d and $r_0 = 0.6\omega_{J,0}$ in panels e–h. Here $\omega_{J,0}/2\pi = 0.47$ THz is the out-of-plane plasma frequency at $T = 0$. In Fig. 3, instead, we set $\gamma_0 = 0.1\omega_{J,0}$, where now $\omega_{J,0}/2\pi = 240$ THz is the $T = 0$ value of the in-plane plasma frequency.

For what concerns the in-plane JPM, we follow the same procedure starting from the interaction term of Eq. (11). In this case, the quartic action has a structure similar to Eq. (18) provided that $K_\perp$ is replaced by a two-component tensor:

$$S_A^{(4)} = \sum_{i\omega_m}A_i^2(i\omega_m)K_{ii;jj}(i\omega_m)A_j^2(-i\omega_m) \tag{20}$$

where $K_{ii;jj}(i\omega_m) = M_{ij}K_\parallel$, and $M_{ij} = \sum_{\mathbf{k}}\frac{k_i^2k_j^2}{\mathbf{k}^4}$. As a consequence, up to an overall normalization factor, one has that $M_{xx} = M_{yy} \propto \int_0^{2\pi}d\phi\cos^4\phi = 3\pi/4$ while $M_{xy} = M_{yx} \propto \int_0^{2\pi}d\phi\cos^2\phi\sin^2\phi = \pi/4$, leading to Eq. (12). If $\theta$ is the angle of the pump field in the plane, i.e., $\mathbf{A} = (A\cos\theta, A\sin\theta, 0)$, then the polarization angle-dependence of the in-plane non-linear optical kernel is[22]:

$$K(\theta) = K_{xx;xx}(\cos^4\theta + \sin^4\theta) + 2K_{xx;yy}\cos^2\theta\sin^2\theta, \tag{21}$$

that can be rewritten in the form of Eq. (14).

**Broadband pump pulse**. For a narrow-band multicycle pulse, one can assume a monochromatic incident field, and the THG is simply related to the non-linear optical kernel via Eq. (8). However, for a broadband pulse with central frequency $\Omega$, the THG is more generally associated with the $3\Omega$ component in the non-linear current[22,23]. We then computed the non-linear current from Eq. (19) by using a realistic pump spectrum $A(\omega)$, obtained by Fourier transform of $A_z(t) = A_0e^{-t^2/\tau^2}\sin(\Omega t)$. The result for $I_z$ is shown in Fig. 2e at different temperatures. The $\tau = 0.85$ ps and $\Omega/2\pi = 0.45$ THz parameters are set in such a way that the e.m. field $E_z(t) \propto -\partial A_z(t)/\partial t$ reproduces accurately the experimental pulse profile of ref. [18].

**Pump-probe configuration**. In a pump-probe experiment designed to excite the out-of-plane JPM, both the pump and probe fields are polarized along $z$, i.e., $E_z = E_{pr}(t) + E_p(t)$. Here, we will refer for simplicity to the transmission configuration, as discussed in ref. [23,37], where one measures the variation $\delta E_{pr}(t)$ of the

transmitted probe field with and without the pump, so that terms not explicitly depending on the pump field cancel out. This allows one to express it as $\delta E_{pr}(t) \propto \int dt' A_z^{pr}(t) K(t-t')(A_z^p)^2(t')$. By considering a fixed $t_g$ acquisition time and implementing the time delay $t_{pp}$ between the pump and the probe, $\delta E_{pr}(t_g; t_{pp})$ becomes a function of $t_{pp}$ only, as given by the first line of Eq. (9). Finally, by computing from Eq. (7) the non-linear kernel in time domain, i.e., $K(t) = \int \frac{d\omega}{2\pi} K(\omega) e^{-i\omega t} = F(T) e^{-\gamma t} \sin(2\omega_J t)$, we derive the last line of Eq. (9). For the reflection geometry used in ref. [17] the basic mechanism is the same, so that one expects that the differential reflectivity signal scales with the convolution of the non-linear kernel times the pump field squared given in Eq. (9). For the simulations in Fig. 2e–h, we used the broadband pump pulse described above. For the in-plane response measured in ref. [21], the huge frequency mismatch between the spectral components of the gauge field and $2\omega_J^{\parallel}$ implies that only the term with $t = t_{pp}$ survives in the integral (9). As a consequence, the oscillations are absent and $\delta E_{pr}(t_{pp})$ simply scales as the square of the pump field, modulated by $F(T)$ and by the polarization encoded in the kernel (12). Indeed, if the pump field forms an angle $\theta$ with the $x$ axis and the probe is applied, e.g., along the $x$ axis, from Eq. (9), properly generalized for the planar configuration, one easily sees that $\delta E_x \sim K_{xx;xx} \cos^2\theta + K_{xx;yy} \sin^2\theta = K_{A_{1g}} + K_{B_{1g}} \cos(2\theta)$. This is exactly the decomposition used to analyzed the transient reflectivity measured in ref. [19].

## Data availability

All data generated during this study are included in this published article (and its supplementary information files).

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

## Acknowledgements

We acknowledge useful discussions with C. Castellani, A. Cavalleri, and D. Nicoletti. We thank the authors of ref. [18] for providing us with the experimental data used to estimate $J_\perp(T)$ in Fig. 2. This work has been supported by the Italian MAECI under the Italian-India collaborative project SUPERTOP-PGR04879, by the Italian MIUR project PRIN 2017 No. 2017Z8TS5B, by Regione Lazio (L.R. 13/08) under project SIMAP and by Sapienza University under project Ateneo 2019 (Grant No. RM11916B56802AFE).

## Author contributions

F.G. and M.U. contributed equally to this work. L.B. conceived the project and supervised its development. F.G., M.U. and L.B. performed the analytical calculations. F.G. and M.U. performed the numerical simulations. L.B. wrote the manuscript with inputs from all the authors.

## Competing interests

The authors declare no competing interests.
