## [Peer Review File · Nature Communications]

REVIEWER COMMENTS

Reviewer #1 (Remarks to the Author):

The manuscript presents a quantum treatment of THz driving of both low-frequency (out-of-plane) and high-frequency (in-plane) plasma waves using a general two-plasmon excitation mechanism. The authors also provide some clearly theoretical predictions to guide the future quantum dynamics of collective modes in unconventional superconductors as well as compare their results with the existing experiments. In my opinion, this is a timely study that addresses some of the most important questions and puzzles in condensed matter physics today. Although THz light pulses have been efficiently used to explore collective modes, the in-plane Josephson plasma mode is still elusive. There are several puzzles for the existing data, especially the persistence of THG signals high above T_c and the potential polarization dependence of A1g and B1g symmetries. The current study establishes the theoretical framework to deal with these problems and offer critical insights into anisotropic JPMs in layered cuprates. The writing style is clean which is suitable to a broad audience. I enjoy reading it and the theoretical formulations are quite assessable to experimentalists. In this sense I like to recommend it for the publication given the following comments and suggestions can be satisfactorily addressed in a revised manuscript.

1. The paper explains the persistence of THG above T_c in cuprate superconductor in Refs [14] and [15] using a generic reason of pronounced phases fluctuations for the in-plane Josephson stiffness at the THz frequency. I would like to request the authors to put the explicit expression of J_{\perp} or fluctuations terms that gives the non-zero signals above T_c , e.g., used to produce Fig. 3c. Can authors also comment the potential doping dependences? In the experiment, it looks like the temperature-dependence THG shape is quite sensitive to the doping which presumably connects to the strength of phase fluctuations. Does the nonlinear THz probe offer a sensitive probe to such doping dependent phase fluctuations? It looks to me the current theory may also shine this very important questions in cuprates and make the current paper more appreciated by a broader community.
2. Recently there has been established a new effect on the lightwave acceleration of supercurrent during the multicycle THz driving, see, e.g., in Physical Review Letters 124, 207003 (2020). Such effects have a direct consequence on dynamic spatial and temporal modulations of the condensate phase. Although the effect is only demonstrated in a BCS superconductor so far, it will be interesting for the authors to comment on influence of such physics to the JPM generation in cuprates or on a potential roadmap to include the cooper pair acceleration during the THz field oscillations in the future theoretical formulation of JPM modes.
3. In order to detect the predicted polarization dependence, what type of samples are recommended, i.e., single crystals vs thin films? Does multi-grain thin film will smear out such polarization dependence? It will be beneficial to add a brief sentence in the revised paper to reach out the experimental community.

Some minor comments:

1. For the sake of completeness of the experimental references on nonlinear THz driving (there is not very much actually in superconductors), I suggest the authors to add a prior literature on discovery of a gapless metastable quasi-particle state by THz driven Anderson pseudospin precessions, Nature Materials. 17, 586 (2018).
 2. At the end of first column and start of second column (page 4), λ_{ac} could be a typo and should be λ_{ab} to me. Please check.
- In summary, this manuscript represent a compelling theoretical study targeting at some of the outstanding problems today. I can recommend it for publication after the above issues addressed.

Reviewer #2 (Remarks to the Author):

The paper by Gabriele et al. deals with layered high-Tc superconductors in non-equilibrium. In addition to the well-known Higgs (amplitude) response of the system the authors investigate the Goldstone (phase) response of these materials in a microscopic way described in Fig.1 after pumping. The out-of-plane response is modeled with parameters taken from experiment (Ref. 6) and thus provides a microscopic justification for some phenomenological models (e.g. sine-Gordon) used earlier. For the in-plane response the authors assumed a preformed-pair scenario (Fig. 3c). For both cases, the Third-Harmonic-Generation (THG) signal is calculated and conclusions can be drawn.

On the other hand the present manuscript has several significant drawbacks. First of all, the manuscript is (except the introduction) not well written, too many technical points are raised without discussing the existing theories on that subject. Even worse: When I read the paper for the first time, I expected an explanation to the experiments in Refs. 7-9 (because of the abstract and motivation), but finally only very little knowledge has been added, see before/after Eq. (14) to these date, but, for example, no explanation of the phase jump in Ref.8 is provided. Is the observed 2nd resonance in Ref. 8 the one at Δ of Eq.(8)?). But later, after reading the manuscript a few times, I believed that the main results are in connection with Ref. 6 and induced superconductivity, even though the results have been used for modelling it. So, not well written.

Secondly, prior to Ref. 6, a well-known scenario for the out-of-plane response follows an interpretation by van der Marel (Uni Geneva) or Bernhard (Uni Fribourg), to name just a few. The authors did not discuss their results in the light of these previous, successful, models. For example: is their resonance at Δ (i.e. inter-layer coupling, I assume) due to tunneling of preformed pairs or real Cooper-pairs? Is the new idea of the plasma mode related to the LC circuit? All previous explanations are not touched. Consequently, the paper is more written for a specialized journal.

Thirdly, the same group published previously other ideas on the interpretation of the THG signal, for example by charge density fluctuations. This means we have in total 3 contributions (Higgs, JPM, CDF) from this group to the THG signal.

What is the relative strength of these contributions? This is not answered by the authors. As it stands they just provide another idea (Fig. 1) and work out some details using Ref. 6. So far, they cannot provide a good explanation for Refs. 7-9 (but this is the expectation the authors raise in the introduction). From a distance this looks like a discussion in public between these groups for years which should not happen in Nature Communications.

In short, this manuscript is suitable for a more specialized journal where this microscopic idea should be published. The heart of the paper, which is written in a technically way, fits better there. As it stands, this manuscript is not for a broader audience. I do not recommend publication in Nature Comm.

R1 Reply to Reviewer 1

We thank the Reviewer for reading our manuscript and for recognizing that it is "a timely study that addresses some of the most important questions and puzzles in condensed matter physics today". We are also glad that the Referee found our writing style "clean, which is suitable to a broad audience". The Reviewer raises some interesting points, that we will discuss in details below.

The paper explains the persistence of THG above T_c in cuprate superconductor in Refs [14] and [15] using a generic reason of pronounced phases fluctuations for the in-plane Josephson stiffness at the THz frequency. I would like to request the authors to put the explicit expression of J_{\perp} or fluctuations terms that gives the non-zero signals above T_c , e.g., used to produce Fig. 3c.

We thank the Reviewer for his/her question, which gives us the opportunity to expand the discussion on the persistence of the THG due to in-plane JPM above T_c . What we suggest in the manuscript is that this effect is intimately linked to the persistence of the superfluid density $J_{\Omega}(T)$ above T_c , where the Ω subscript here highlights the fact that what matters is the superfluid stiffness measured at a finite frequency Ω . This is determined experimentally as $J_{\Omega}(T) = \sigma_2(\Omega, T)/\Omega$, and it can differ in general from the real phase rigidity, defined as the limit for $\Omega \rightarrow 0$, i.e. $J(T) = \lim_{\Omega \rightarrow 0} \sigma_2(\Omega, T)/\Omega$. Here $\sigma_2(\Omega, T)$ denotes the imaginary part of the optical conductivity measured at the THz frequency Ω at which the THG experiment is carried out. In any superconductor, both in conventional ones (see e.g. M. Mondal et al. Sc. Rep. 3, 1357 (2013)) and in cuprates (see Refs. [21,39,40] in the manuscript) $J(T)$ vanishes exactly at T_c , but the phase rigidity measured at the finite length scale set up by the finite frequency Ω of the experiment survives in general in a certain range of temperatures above T_c . This finding can be ascribed both to SC fluctuations above T_c and to intrinsic inhomogeneity of the SC properties, with a possible cooperative effect of both phenomena. In the case of Gaussian (amplitude and phase) fluctuations the intrinsic mechanism relies on the fact that preformed Cooper pairs appear bounded already above T_c at the finite length scale set by the probe frequency, leading to a superfluid response. The range of temperature above T_c where this effect is appreciable depends on several factors: even in a conventional BCS superconductor as NbN the above mentioned work has shown an enhancement of the fluctuation regime as the system approaches the superconductor-to-insulator transition getting intrinsically inhomogeneous. A second mechanism possibly responsible for the survival of $J_{\Omega}(T)$ above T_c can be linked to phase fluctuations only, which are expected to be enhanced in cuprates due to the quasi-two-dimensional nature of the pairing.

The full microscopic understanding of this issue is still nowadays an open problem, whose explanation goes beyond the scope of our manuscript. Nonetheless, we point out that as soon as J_{Ω} persists above T_c , also the THG signal is predicted to persist. For the matter of a qualitative comparison with the experiments, we then considered the case where a finite $J_{\Omega}(T)$ above T_c can be ascribed to emergent inhomogeneity of the superconducting ground state. **In the new Sec. S3 of the supplementary file we include a detailed discussion of the model used to generate tails in the superfluid stiffness above T_c , and we briefly mention this issue in the revised version of the main text.**

Can authors also comment the potential doping dependences? In the experiment, it looks

like the temperature-dependence THG shape is quite sensitive to the doping which presumably connects to the strength of phase fluctuations. Does the nonlinear THz probe offer a sensitive probe to such doping dependent phase fluctuations? It looks to me the current theory may also shine this very important questions in cuprates and make the current paper more appreciated by a broader community.

We thank the Reviewer for the question about the doping dependence. As he/she correctly states, within a XY model picture the survival of the THG signal above T_c and its overall strength scale with the strength of the superfluid stiffness. The doping dependence of $J(T = 0)$ has been widely studied in cuprates, and it has been shown that it has a dome-like shape, with a maximum around optimal doping. As a consequence, a detailed experimental investigation of the doping dependence of the THG in cuprates offers certainly a new knob to address the manifestation of phase-fluctuations effects in cuprates. **In the revised version we comment more extensively on this issue, also in relation to the relative strength of different effects, as asked by Reviewer 2.**

Recently there has been established a new effect on the lightwave acceleration of supercurrent during the multicycle THz driving, see, e.g., in Physical Review Letters 124, 207003 (2020). Such effects have a direct consequence on dynamic spatial and temporal modulations of the condensate phase. Although the effect is only demonstrated in a BCS superconductor so far, it will be interesting for the authors to comment on influence of such physics to the JPM generation in cuprates or on a potential roadmap to include the cooper pair acceleration during the THz field oscillations in the future theoretical formulation of JPM modes.

We thank the Referee for pointing out the connection to this very interesting work from the group of J. Wang. This is indeed a remarkable experiment where the authors show the possibility to generate second harmonics of the incoming pump field. This finding is attributed to the emergence of a finite dc component of the local field inside the superconductor, due to the non-linearity itself of the local response. The Reviewer is also right that we did not consider such a possibility in our calculations, and that it is certainly interesting to explore the consequences of it for future work. Indeed, on very general ground, adding this effect could make possible to excite a *single* plasma mode, since its momentum will be compensated by the momentum of the induced supercurrent. **In the revised version of the manuscript we mentioned also this manuscript, along with a second one from the same group we already included (Ref.s [31,41]).**

In order to detect the predicted polarization dependence, what type of samples are recommended, i.e., single crystals vs thin films? Does multi-grain thin film will smear out such polarization dependence? It will be beneficial to add a brief sentence in the revised paper to reach out the experimental community.

At the best of our knowledge, direct THG measured in transmission experiments requires to use thin films in order to have enough transmitted signal, see indeed Ref. [20]. On the other hand, pump-probe spectroscopy uses the reflected signal to analyse the polarization dependence and it can be also carried out in single crystals, as one in Ref. [19]. Of course, in both cases one needs single-domain systems in order to avoid to average out the polarization dependence of the signal coming out from grains with different orientation. **In the revised version we added a short sentence on this aspect.**

For the sake of completeness of the experimental references on nonlinear THz driving (there is not very much actually in superconductors), I suggest the authors to add a prior literature on discovery of a gapless metastable quasi-particle state by THz driven Anderson pseudospin precessions, Nature Materials. 17, 586 (2018).

We thank the Reviewer for pointing out this reference. This is indeed an interesting work, since it shows that a very intense pump field (about one order of magnitude than the one used by Shimano's group) drives a different non-equilibrium phenomenon, not simply captured by the present perturbative approach. **In the revised version we mention this issue explicitly, and we added the suggested work as Ref. [36].**

At the end of first column and start of second column (page 4), λ_{ac} could be a typo and should be λ_{ab} to me. Please check.

The Reviewer is perfectly right, we thank him/her for pointing out this misprint, that has been corrected in the revised version.

In summary, this manuscript represent a compelling theoretical study targeting at some of the outstanding problems today. I can recommend it for publication after the above issues addressed.

We thank the Reviewer for his/her insightful comments, and we hope he/she can find our revised version appropriate for publication.

R2 Reply to Reviewer 2

We thank the Reviewer for the careful reading of the manuscript and for raising critical comments. We agree with the Reviewer that Nature Communication should aim at a broad audience, and we are grateful for any comment that can help us making the presentation appropriate for it. Below we provide a point-to-point answer to all Reviewer's comments, and we outline the modifications introduced in the manuscript to account for them.

On the other hand the present manuscript has several significant drawbacks. First of all, the manuscript is (except the introduction) not well written, too many technical points are raised without discussing the existing theories on that subject. Even worse: When I read the paper for the first time, I expected an explanation to the experiments in Refs. 14-16 (because of the abstract and motivation), but finally only very little knowledge has been added, see before/after Eq. (14) to these date, but, for example, no explanation of the phase jump in Ref.15 is provided. Is the observed 2nd resonance in Ref. 15 the one at Delta of Eq.(8)?). But later, after reading the manuscript a few times, I believed that the main results are in connection with Ref. 12 and 13 and induced superconductivity, even though the results have been used for modelling it. So, not well written.

We regret that the Reviewer found the presentation of the theoretical part unsatisfactory. Let us first clarify the main aim of the present manuscript. As we state in the abstract, we explore the contribution of two-plasmon excitations to THG. In the case of cuprates the strong anisotropy of the superfluid stiffness for out-of-plane or in-plane response leads to a completely different response in the two cases. In cuprates, experiments have been carried out in both configurations, and we believe that both cases are interesting to explore. Refs. [12-13] (Ref.s

[17-18] in the revised version) report results for the out-of-plane response, Refs [14-16] (Ref.s [19-21] in the revised version) for the in-plane one. The paper is aimed at discussing **both** cases, not only refs. [14-16]. **To avoid any possible misunderstanding, we stressed it more clearly in the revised introduction.** As a logical way of proceeding, in the paper we first discuss the case of the out-of-plane response, ref. [12-13] where no debate exists in the literature about the prominent role of the plasmon, and then we discuss the implications for the in-plane one. Notice that **we never mention the issue of the induced superconductivity**, that has nothing to do with THG, so it is not evident to us why the Referee had this impression. Finally, for the in-plane response we mention three experimental papers, Ref.s [14-16], where several puzzling results have been reported: (i) the temperature dependence of the THG, with a persistence above T_c ; (ii) its polarization dependence and (iii) an observed phase shift between 1st harmonic and 3rd harmonic. While the items (i)-(ii) are common to all three manuscripts, item (iii) has only been reported in Ref. [15]. To address this issue, one must properly model both the linear and non-linear current transmitted through the sample, which also requires to add details of the sample. For this reason, we decided not to address this issue, and we never mention it, neither in the abstract nor in the introduction. In contrast, we offer a plausible explanation for items (i)-(ii), which holds regardless the specific characteristic of the experimental configuration of Ref. [15]. **To avoid any possible misunderstanding, we detailed more clearly in the revised manuscript the features under scrutiny for what concerns the in-plane response, both in the introductory paragraph and in the discussion of the in-plane response.**

Secondly, prior to Ref. [12] and [13], a well-known scenario for the out-of-plane response follows an interpretation by van der Marel (Uni Geneva) or Bernhard (Uni Fribourg), to name just a few. The authors did not discuss their results in the light of these previous, successful, models.

We agree with the Referee that the existence of an out-of-plane soft plasma mode in the SC state of cuprates was highlighted by standard spectroscopic measurements back in the 90thies by several experimental groups. However, all this previous work focuses on the *linear* optical response, and indeed standard reflectivity measurements had been used to evidence the plasma edge in the SC state. What is new in Refs. [12]-[13] is the use of intense THz pulses in order to trigger a *non-linear* optical response, and to measure its effects via either pump-probe protocols or third-harmonic generation. On the other hand, these effects have been discussed in the literature (see Refs. [11-16]) using as starting point exactly the layered Josephson model originally proposed to explain also linear spectroscopy. **For the sake of completeness, we added in the revised introduction the explicit reference to prior results of standard spectroscopy aimed at detecting and explaining the existence of a soft out-of-plane plasma mode, see Refs. [6-10].** We included a list of highly-cited papers on this issue, but since the literature on the subject is rather wide we could have missed some relevant one. We will be glad to include it if the Reviewer can provide us with a precise article citation.

For example: is their resonance at Delta (i.e. inter-layer coupling, I assume) due to tunneling of preformed pairs or real Cooper-pairs? Is the new idea of the plasma mode related to the LC circuit? All previous explanations are not touched. Consequently, the paper is more written for a specialized journal.

Since we discuss the plasma modes the relevant energy scale is the phase stiffness, which is the energetic cost to create a phase gradient in the SC state. The starting Eq. (1) cannot be

misinterpreted: this is a standard Josephson model, with inter-layer Josephson coupling J_{\perp} , as stated in the text. We believe that the Josephson model is so well known in the literature that such a starting point cannot be considered appropriate for a specialized journal. The symbol Δ is used in the manuscript to identify the SC gap, but since it does not enter the superfluid response, in our figures 2,3 there is no resonance at Δ . We would be glad to answer the Reviewer's question, but its present formulation does not allow to do so.

Thirdly, the same group published previously other ideas on the interpretation of the THG signal, for example by charge density fluctuations. This means we have in total 3 contributions (Higgs, JPM, CDF) from this group to the THG signal. What is the relative strength of these contributions? This is not answered by the authors. As it stands they just provide another idea (Fig. 1) and work out some details using Ref. 12 and 13. So far, they cannot provide a good explanation for Refs. 14-16 (but this is the expectation the authors raise in the introduction).

We thank the Referee for raising this important point, which gives us the opportunity to clarify the current state of the art. The first experiments on THG in a conventional NbN superconductor by the group of Shimano, Ref. [28], showed that the non-linear optical kernel has a resonance at 2Δ . By making a calculation within a **clean** BCS model the authors attributed this result to the Higgs mode. Subsequent theoretical work by our group, Ref. [22], showed that that theoretical interpretation was unfortunately wrong, since within a clean BCS model the Higgs signal is largely subleading with respect to lattice-modulated charge density fluctuations (CDF). At the same time we highlighted the fact that the two contributions can be distinguished not only by their relative intensity, but also by their polarization dependence. However, new experimental work triggered by our results evidenced that in the experiments in conventional NbN the polarization dependence was not the one expected from CDF. Triggered by these findings, some authors, see Ref. [24]-[25] addressed the role of **disorder**, and they showed that within a disordered BCS model the nature and relative strength of quasiparticle and Higgs contribution changes, and ultimately for strongly disordered conventional superconductors the Higgs actually dominates. All these results are beautifully reviewed in Ref. [26], and essentially a general consensus has been reached so far on the relevance of the Higgs mode for the case of films of NbN. These findings were already briefly mentioned in the previous version.

On the other hand, when the first experiments for an in-plane pump field on cuprates came out, new experimental results were observed, not easily understood on the basis of existing theories. This is not surprising: cuprates have a layered structure, with anisotropic response, and are relatively clean unconventional superconductors, with a plethora of effects not seen in conventional NbN. It is then natural that also new theoretical proposals are needed, aimed at highlighting other effects, not discussed so far, that can be relevant for these specific materials. For example, due to the incoherent nature of transport along the planes one does not expect any relevance of quasiparticle and Higgs processes for a pump field polarized perpendicular to the planes. And indeed, also theoretical work previous to ours focused only on the plasmon excitations. At the same time, the new findings for the pump field in the plane, i.e. the scaling of the THG with the stiffness and its polarization dependence (see items (i) and (ii) above), are not accounted by existing theoretical results. These are the features discussed in our manuscript.

In summary, the Referee is right that several processes can be relevant to explain the non-linear optical response in cuprates: the quasiparticles, the Higgs mode and the two-plasmon processes. They can be distinguished by the resonance frequency, the polarization dependence and the relative strength. In the present work we derive on general grounds the expected temperature and polarization dependence of the plasmon contribution for both out-of-plane and

in-plane response. For the out-of-plane we provide results in very good quantitative agreement with existing data. For the in-plane one we show a new mechanisms able to explain the temperature and polarization dependence. However, as already mentioned in the manuscript, to establish the relative strength of the two-plasmon contribution with respect to Higgs and quasiparticles, one needs to go beyond RPA. In addition, in accordance to recent theoretical work, this has to be done within a disordered microscopic model. This is a formidable task. As two of us have shown in a recent preprint, Ref. [27], already accounting properly for the strength and polarization dependence of the quasiparticle and Higgs at RPA level is rather challenging, and it requires to implement a full numerical solution. On the other hand, a result of Ref. [27] is that these two contributions cannot account for the in-plane mixed $A_{1g} + B_{1g}$ response of cuprates, while our manuscript show that plasmon processes can explain it. As usual, theoretical work is done in specific models, and with a certain degrees of approximation. It is the comparison with experiments that provides guidelines to decide what is the most appropriate for any specific system, so it is natural that new experiments stimulate new theoretical developments.

In order to account for the Reviewer's comment, we revised several paragraphs of the manuscript in order to give a proper discussion of other effects - besides the plasmons - contributing to the THG, and clarifying the role of disorder to change the hierarchy of the various contributions. We particularly emphasise how they can be distinguished not only by their relative strength, but also by their temperature and polarization dependence.

From a distance this looks like a discussion in public between these groups for years which should not happen in Nature Communications. In short, this manuscript is suitable for a more specialized journal where this microscopic idea should be published. The heart of the paper, which is written in a technically way, fits better there. As it stands, this manuscript is not for a broader audience. I do not recommend publication in Nature Comm.

We gently disagree with the Reviewer on this point. It is true that there has been a scientific debate in the literature about the relative role of several effects, but so far this stimulated an intense experimental and theoretical work which contributed significantly to improve our understanding of the non-linear response of superconductors. This is also the ultimate goal of this manuscript, and due to the wide interest that this issue attracted in the recent literature we believe that it is important to publish these results in a wide context as the one offered by Nature Communications. We also think that the useful suggestions given by both Reviewers helped us to make the presentation more accessible to a broad audience.

REVIEWER COMMENTS

Reviewer #1 (Remarks to the Author):

The authors have done a comprehensive and satisfactory job to address my comments. Their efforts have improved the manuscript notably. As I assessed in my prior review, the manuscript presents a much needed treatment of THz driving nonlinear responses, especially in the context of plasma waves and persisting SC fluctuations, in unconventional superconductors. This work therefore provides some clear theoretical predictions to guide the search for new phase and amplitude modes in the future. For the final note, I would like to point out a new experimental reference on THz driving of amplitude modes in unconventional superconductors (Vaswani, et al., ArXiv: 2011.13036).

In summary, this manuscript represents a compelling theoretical study of phase and amplitude modes in the nonlinear THz responses. I recommend the revised manuscript for publication.

Reviewer #2 (Remarks to the Author):

The authors have addressed most of my questions:

- a) revised introduction is very much improved
- b) including of citations of previous approaches (e.g. Geneva group)
- c) LC circuit: new Ref. 10 is OK
- d) several paragraphs revised concerning the role of Higgs vs JPR vs CDF (with impurities)

On the other hand I am still not very much impressed by the main result that the anisotropy of the Josephson couplings leads to marked differences in the thermal effects among the out-of-plane and in-plane response.

In a phonon-based superconductor (e.g. NbSe₂) a corresponding CDW should also produce a THG signal. In multiband Fe-based superconductors the corresponding Badaš-Schrieffer mode will also produce a THG signal and so on. Of course, in layered cuprates the plasma frequency is shifted down and becomes anisotropic: both branches will produce a THG signal as well, no surprise. Other groups (e.g. PR Research 2, 023413 (2020)) have shown that more modes will be important and how they couple, also acoustic plasmons in bilayer systems and so on.

If the authors now claim that only the T-dependence is the a key result in this field, then one should explain also the phase jump in Ref. 15. Otherwise the insight of this manuscript is limited.

It is a close call.

Reviewer #3 (Remarks to the Author):

The authors present a calculation of a new contribution to the THz non-linear optical signal in the cuprates; the Josephson plasmon contribution. They do so by deriving low-energy effective field theories for the plasmon.

The third-harmonic response of superconductors has been of some interest in the community of researchers working on light-induced phenomena and non-linear optical properties of superconductors.

By their nature, non-linear processes are difficult to interpret, derive, and discuss. Owing to the presence of multiple fields, the number of experimental options and of contributions to a particular signal grows rapidly.

This situation has (in part) led to the current efforts of these authors and others to suss out the nature of the third harmonic signal. This particular work has identified a new contribution — a two-plasmon excitation (similar to e.g. a two-magnon or 2-phonon excitation in Raman scattering).

The complications also result in the relatively dense mathematics in the manuscript (or at least more dense than is typically shown). Yet, I find that the level of detail is necessary and enlightening.

While Referee B is correct that this is part of a discussion between two groups that has been going for some years, that is not a reason to discount results. The referee is, however, correct in their assessment that now several mechanisms exist, this is yet another one, and we do not have a clear idea of the relative strengths (although the authors' comment regarding the contribution in a unique channel mitigates this somewhat).

Overall, non-linear phenomena are on the rise in the level of interest from the community as a whole. I believe this paper makes a valuable contribution, and can be published in Nature Communications after some further issues are addressed.

Most notably, the manuscript is somewhat difficult to follow. The text is dense and one topic flows into another without (always) a clear transition or break.

* For readability, the manuscript could benefit from some organizational structure. The authors address several different scenarios, and a simple heading would suffice to guide the reader. Additional paragraph breaks would also be beneficial.

* The readability has some further issues; for example, in the second paragraph the authors have a "We first..." statement, with no following "Next" or similar. This makes it difficult to follow.

Aside from readability, I have a few technical questions:

* The authors make reference to strong and weak THz fields. An estimate of peak field would be beneficial.

* The NL current has a strong response in the first harmonic channel as well, which is not mentioned (as far as I can tell).

* What is the origin of the non-monotonicity in Fig 2f?

* What is the NL current in Fig 2 normalized to? Is the normalization the same across panels?

* Below Eq. 5 the authors use the symbol S_G , which has not been defined (presumably a \perp is

missing, which may also be the case for K further down).

* Eqs (2) and (3) would benefit from time indices

* The variation of the action with respect to the field A_z should yield two contributions (both time coordinates are integrated over). If the authors collapsed these into one in the expression for the NL current (below Eq 7) via use of symmetries in $K(t-t')$, this should be made clear. Otherwise, an additional line or two of derivation would be beneficial.

R1 Reply to Reviewer 1

The authors have done a comprehensive and satisfactory job to address my comments. Their efforts have improved the manuscript notably. As I assessed in my prior review, the manuscript presents a much needed treatment of THz driving nonlinear responses, especially in the context of plasma waves and persisting SC fluctuations, in unconventional superconductors. This work therefore provides some clear theoretical predictions to guide the search for new phase and amplitude modes in the future. For the final note, I would like to point out a new experimental reference on THz driving of amplitude modes in unconventional superconductors (Vaswani, et al., ArXiv: 2011.13036).

In summary, this manuscript represents a compelling theoretical study of phase and amplitude modes in the nonlinear THz responses. I recommend the revised manuscript for publication.

We thank the Reviewer for reading again our manuscript, and for pointing out a new reference that appeared after our resubmission. This is indeed an interesting work on non-linear response of pnictides. We added the citation to this work as Ref. [33] in our revised manuscript. We are very glad that, considered all revisions, Reviewer 1 recommended our paper for publication in Nature Communications.

R2 Reply to Reviewer 2

The authors have addressed most of my questions: a) revised introduction is very much improved b) including of citations of previous approaches (e.g. Geneva group) c) LC circuit: new Ref. 10 is OK d) several paragraphs revised concerning the role of Higgs vs JPR vs CDF (with impurities)

On the other hand I am still not very much impressed by the main result that the anisotropy of the Josephson couplings leads to marked differences in the thermal effects among the out-of-plane and in-plane response.

We thank the Reviewer for reading again our manuscript, and for appreciating our efforts to improve it according to his/her suggestions. We are slightly puzzled by his/her comment: of course, any new result is obvious after it has been demonstrated. However, as far as we know nobody discussed the role of two-plasmon excitations for the in-plane response of cuprates, and the current understanding of the out-of-plane response was limited to a semiclassical approach. In addition, we also highlight the different polarization dependence of the two-plasmon signal with respect to the Higgs/quasiparticles one, which is something to be tested experimentally. All these results are definitively new and timely, as the other two reviewers recognized.

In a phonon-based superconductor (e.g. NbSe₂) a corresponding CDW should also produce a THG signal. In multiband Fe-based superconductors the corresponding Badasis-Schrieffer mode will also produce a THG signal and so on. Of course, in layered cuprates the plasma frequency is shifted down and becomes anisotropic: both branches will produce a THG signal as well, no surprise. Other groups (e.g. PR Research 2, 023413 (2020)) have shown that more modes will be important and how they couple, also acoustic plasmons in bilayer systems and so on.

If the authors now claim that only the T-dependence is the a key result in this field, then

one should explain also the phase jump in Ref. 15. Otherwise the insight of this manuscript is limited. It is a close call.

The reviewer raises an interesting point: any Raman-active mode can contribute to the non-linear response, so what's really new here? Actually, all the recent work done by our group and others within the context of non-linear THz spectroscopy shows that actually this is not true: there is a difference between the Raman response - intended as visible-light scattering - and the one measured with THz light. Indeed, in the latter case the microscopic response involves paramagnetic processes that are activated by disorder but are *absent* in the "standard" Raman response, which maps out everything in a diamagnetic-like response. Just to be clear, the paper mentioned by the Reviewer, that was already cited in our manuscript as Ref. [32], is just a summary of (partly well-known) results concerning the collective-modes contribution to the diamagnetic density-density response. They can be straightforwardly applied to THz spectroscopy in the limit of extremely clean superconductors. This is a point not very clear yet in the community. In other words, the THz non-linear kernel has selection rules *different* with respect to ordinary Raman spectroscopy. This is the real breakthrough of these experiments, that has now become clear thanks to the intense theoretical efforts by many groups. Our present manuscript just adds one more piece to the puzzle. We agree that the puzzle is not complete yet, and other features like the phase jump of Ref. 15 should be addressed. But as we already said, this cannot be an argument to discount our results.

R3 Reply to Reviewer 3

We thank the Reviewer for reading our manuscript and all the previous correspondence. We are glad that the Reviewer, after taking into account carefully the criticisms raised by reviewer 2, suggested anyway publication in Nature Communications after some revision. We provide below a point-to-point reply to all the Reviewer's comments.

Most notably, the manuscript is somewhat difficult to follow. The text is dense and one topic flows into another without (always) a clear transition or break. For readability, the manuscript could benefit from some organizational structure. The authors address several different scenarios, and a simple heading would suffice to guide the reader. Additional paragraph breaks would also be beneficial. The readability has some further issues; for example, in the second paragraph the authors have a "We first.." statement, with no following "Next" or similar. This makes it difficult to follow.

We thank the Reviewer for suggesting this revision, that we are glad to implement. We are perfectly conscious that the manuscript is rather dense. On the other hand, we think that it is important to retain at least the main steps of the theoretical derivation. In order to separate more clearly the various sections, and in particular the results for the two different polarization (out-of-plane and in-plane) we included section headings in the revised version, and we clarified better the logical sequence of the derivation in the introductory paragraph.

The authors make reference to strong and weak THz fields. An estimate of peak field would be beneficial.

According to the current status of experiments in superconductors, the typical strength of the field where a non-linear "quasi-equilibrium" response is still valid is around 100 kV/cm. For larger fields one can eventually melt the superconductors and induce a rather different dynamics, as observed for example in Ref. [36]. We added this estimate in the revised version.

The NL current has a strong response in the first harmonic channel as well, which is not mentioned (as far as I can tell).

The Reviewer is right. The non-linear current has a first harmonic as well, and it also scales with the non-linear kernel. For a monochromatic pump field at Ω it goes like $2K(0) + K(2\Omega)$. However, in transmission experiments the first harmonic can also originate from the linear response, that is not completely screened in the superconductor at finite temperature. For this reason, it is hard to get an information on K only from the first harmonic. In the revised version we added a short comment on this before than Eq. (8).

What is the origin of the non-monotonicity in Fig 2f?

We thank the Reviewer for raising this point, that was probably not clear enough in the previous version. In general, the THG for the out-of-plane response is not monotonic, and what is seen in Fig. 2f is the broad-band analogous of what found for the narrow-band case in Figs 2b,d. Indeed, one has to face with three different temperature effects in the kernel of Eq. (7) evaluated at 2ω : (i) the suppression of $J_{\perp}(T)$ and $\omega_J(T)$ with temperature; (ii) the increasing of $\coth(\beta\omega_J)$ with temperature due to thermal activation of the plasmon population; (iii) the resonance condition $2\omega = 2\omega_J(T)$, that is achieved at the temperature where the (fixed) pump frequency matches the value of ω_J . Depending on the value of $\omega/\omega_J(T=0)$ and of the damping, for a monochromatic pump field one has the three different curves of Figs 2b,d. The case of Fig. 2f corresponds more or less to the case $\omega = \omega_2$ in panel 2d. However, since in this case we simulate the broad-band pulse used in the experiments of Ref. [18] the overall shape is slightly different from the one in Fig. 2f. In the previous version these effects were discussed, but probably we failed to convey the message in a clear way. In the revised version we modified the paragraph where we explain the temperature dependence of the THG for out-of-plane response, and we also highlight in the introduction the difference among in-plane and out-of-plane response.

What is the NL current in Fig 2 normalized to? Is the normalization the same across panels?

We thank the Reviewer for raising this point, which deserves further clarification. We used actually different normalization for the non-linear current in the various panels in Fig. 2. In panels b,d we normalized the I^{THG} to the $T = 0$ value for the ω_2 pumping frequency. This choice simply follows from the need to show in the same panel results for different pumping frequencies, where the resonance occurs at different temperatures. The data shown in panel 2f are derived by integrating the peak around $\omega = 3\Omega$ in the nonlinear current shown in panel 3e (grey region) at different temperatures. In this case I^{THG} is normalized to its $T = 0$ value. We have better clarified these normalizations in the revised version of the manuscript.

Below Eq. 5 the authors use the symbol S_G , which has not been defined (presumably a \perp is missing, which may also be the case for K further down).

We thank the Reviewer for pointing out these misprints, that have been corrected in the revised version

Eqs (2) and (3) would benefit from time indices

We agree with the Reviewer that in Eq. (2) time can be easily added, and we did it in the revised version. For what concerns Eq. (3), at this level the expression is rather general, so the time convolution depends on the nature of the non-linear processes. In other words, while it is easy for Kubo-like diagrams as the one shown in Fig. 1b, it is more involved for any general fourth-order process. For this reason, we decided to omit the explicit time dependence in Eq. (3).

The variation of the action with respect to the field A_z should yield two contributions (both time coordinates are integrated over). If the authors collapsed these into one in the expression for the NL current (below Eq 7) via use of symmetries in $K(t - t')$, this should be made clear. Otherwise, an additional line or two of derivation would be beneficial.

We thank the Reviewer for pointing out this misprint. It was irrelevant for the overall results, since the current is always normalized to an arbitrary prefactor, but the Reviewer is right that actually a factor of 4 is missing. Taking advantage of the Reviewer question we decided to uniform the notation, using imaginary time and Matsubara frequencies in all the equations (4)-(6), and we only introduce the real frequencies after analytical continuation in Eq. (7). In the revised Method section we also provide more details on the derivation of the non-linear current, explaining the symmetry of the kernel used to get the overall prefactor.